# The "Walking Estimated Limitation Stated by History" (WELSH) visual tool is applicable and accurate to determine walking capacity, even in people with low literacy level

**Wendsèndaté Yves Sempore**[1,2,3ʘ], **Nafi Ouedraogo**[4ʘ], **Salifou Gandema**[5], **Samir Henni**[2], **Alassane Ilboudo**[6], **Téné Marceline Yameogo**[6], **Pierre Abraham**[2,3]*

1 Centre Muraz, Institut National de Santé Publique, Bobo Dioulasso, Burkina Faso, 2 Vascular Medicine Department, Centre Hospitalier Universitaire d'Angers, Angers, France, 3 UMR CNRS 6015, INSERM 1083, Institut MitoVasc, Université d'Angers, Angers, France, 4 Physiology, Centre Hospitalier Universitaire Sourô Sanou, Bobo Dioulasso, Burkina Faso, 5 Physical Medicine and Functional Rehabilitation Departement, Centre Hospitalier Universitaire Sourô Sanou, Bobo Dioulasso, Burkina Faso, 6 Department of Internal Medicine, Centre Hospitalier Universitaire Sourô Sanou, Bobo Dioulasso, Burkina Faso

ʘ These authors contributed equally to this work.
* piabraham@chu-angers.fr

## Abstract

Determination of the self-reported walking capacity by interview or standardized questionnaire is important. However, the existing questionnaires require the patient to be able to read and write in a specific language. We recently proposed the WELSH (Walking Estimated Limitation Stated by History) tool to be administrable to illiterate people. The main objective was to assess the applicability of WELSH tool in the community and in a large group. We performed a prospective study in the city of Bobo-Dioulasso in Burkina Faso during June 2020. We recruited 630 interviewers among medical students. They were trained to administer the WELSH, and to conduct a 6-minute walk test. We performed a Pearson's "r" correlation between the WELSH and maximal walking distance (MWD). Of the 1723 participants available for the analysis, 757 (43.9%: 41.6–46.3) never went to school or attended only elementary school. The percentage of questionnaires with participant filling-in errors corrected by the investigator decreased with the decrease in educational level ($p<0.001$). The average WELSH score was 53 ± 22 and the average MWD was 383 ±142 meters. The Spearman correlation coefficient between the WELSH score and the MWD was r = 0.567 ($p<0.001$). Correlations ranged from 0.291 to 0.576 in males and females, (all p values < 0.05) and in different levels of education, with the highest coefficients found in illiterate people. The WELSH is feasible on the community by a wide variety of interviewers. It correlates with the MWD estimated by the 6-minutes' walk test even for people with little or no schooling.

**Data Availability Statement:** All relevant data are within the paper and its Supporting Information files.

**Funding:** The author(s) received no specific funding for this work.

**Competing interests:** The authors have declared that no competing interests exist.

## Introduction

The assessment of functional walking ability often constitutes a decision-making element in therapeutic indications and patient follow-up [1–4]. Determination of the self-reported walking capacity by interview or standardized questionnaire is important both because it facilitates epidemiological studies and because it reflects patients perception of their physical impairment [5]. One of the most widely used questionnaires that has been proposed to standardize the subjective assessment of walking impairment is the "Walking Impaired Questionnaire" (WIQ) [1,6]. Unfortunately, the WIQ is relatively long to fill and almost impossible to score by mental calculation [7]. Our group developed the "Walking Estimated Limitation Calculated by History" (WELCH) to solve some of these issues [2,5,8]. The WELCH consists in three questions on the maximal time (and not distance) that a task can be performed for three different paces and one question on the usual walking pace. The assumption behind this concept is that time is easier to determine than distance, and that time decreases if pace increases. Scoring of the WELCH is based on adding the scores for estimated times and multiplying the result of this addition by a coefficient attributed to each possible usual walking pace. The WELCH has been validated in various languages [9–11]. However, as for any of the questionnaires available to date, questionnaire filling requires that the patient and/or the health worker administering the test can read and write in a specific language. This makes it difficult to apply questionnaires in developing countries with low literacy rates, while walking remains an important means of travelling and working in such countries. We recently proposed the Walking Estimated Limitation Stated by History (WELSH) as an adapted version of the WELCH (and not a translation of the WELCH into images) that aims to be administrable to illiterate people. The WELSH aimed to keep the concept of time estimation at difference paces as well as analysis of usual walking pace. It should be noted that while the durations proposed for the WELCH followed an exponential increase, the concept was not easily transferable to the non-literate WELSH, where the clock was divided in simple intervals to facilitate the scoring. Similarly, aiming to simplify the WELSH, and to remain as pragmatic as possible, the scoring of the WELSH was defined arbitrarily and built to be easily memorized and to be calculated by mental calculation by the users. We previously demonstrated that the WELSH correlated to measured walking impairment in a small group [12]. The main objective of the present study was to assess the usability of the WELSH by non-expert users, its applicability in the community and in a large group and estimate whether it could be used even in people with low level of literacy to estimate MWD.

## Materials and methods

We conducted a prospective study during the month of June 2020 in the city of Bobo Dioulasso in Burkina Faso. Inclusion criteria for participants were age of 20 years old or more, understanding and agreement to participate in the study, ability to read the time on a watch, ability to perform a 6-minute walking test on a flat surface. We did not include individuals who were unable to walk or had unstable chronic disease or a history of recent (< 3 months) acute disease. Since our aim was to test whether the WELSH could be used as a routine tool, we did not want to have the experiment performed by senior physicians but rather by naïve non expert students. Then, among second year medical students of University Nazi BONI, we recruited 630 interviewers. All interviewers were explained the study, shown the documents and material of the study, trained to administer the WELSH and medical interview, detailed inclusion and exclusion criteria of the study, explained which explanation should be proposed to the participants and shown how to perform a 6-minute walk test. Then, all interviewers performed a training session by completing the questionnaire and doing a 6-minute test to one of

the other interviewers, by groups of two students under senior supervision. Each interviewer was provided three papers printed with the WELSH on one side and the parameters of clinical characteristics to be recorded on the other side. Interviewers were asked to use their personal watch for time measurement. For each participant, we collected age, sex, measured or self-reported weight, height (from the identity card), level of education and chronic morbid conditions, if any. Thereafter, groups of two students/interviewers were constituted to recruit 6 participants per group among their neighbors, relatives (each interviewer was asked to include three participants). Each group was given a 30 meters long rope with plastic cones at each end and was given two weeks to recruit their participants and return all their completed files.

## Ethical consideration

The study protocol was approved by the institutional ethics committee of the MURAZ Center under the number 2020-01/MS/SG/INSP/DG/CEI of February 03, 2020, and was registered on clinicaltrials.gov with the identifier NCT03482869. It was performed according to the International Ethics Standards and conforms to the Helsinki Declaration. For each participant after oral information of the study goal in the participant's language or dialect, a signature confirming informed consent to participate in the study was obtained from all.

## Questionnaire design and completion

The WELSH is a visual tool that contains 4 items and has been previously reported [12]. In brief, for the first three items, the maximum walking time that can be performed for each of 3 different walking speeds (illustrated by a turtle a human and a rabbit) must be reported. Walking speeds are considered relative to the people of the same age, family or friends. Each participant was asked to mark the estimated time by a pencil on the image of the dial of a pointer watch (with a mark on the clock: Fig 1). A score from 0 to 7 is assigned to each item depending on the number of minutes estimated by the participant. Note that the animals were chosen on purpose to exist on all 5 continents. For example, the rabbit was preferred to the antelope that is not present on, all continents. Details of scoring method are presented in Fig 1. For the three first items, the number of points increased by one point for each interval of five minutes up to 20 minutes and for each interval of 10 minutes from more than 20 minutes.

For the fourth item, the participants were asked to estimate their usual walking pace as being much slower (Snail), slower (Turtle), similar (Human), or faster (Rabbit) than people of the same age, family or friends by circling the adequate image in their usual daily activities. An estimated usual walking speed is attributed coefficients ranging from one for snail to four for rabbit' (Fig 2).

The score is calculated by adding one to the sum of the points obtained on the first 3 items, and multiplying the result of the addition by the coefficient obtained on the 4th item. The resulting WELSH score ranges from one (lowest capacity, severe impairment), to hundred (absence of impairment) (Fig 3).

During oral explanation in the participant's language or dialect of how to complete the questionnaire the participants, the interviewers underlined that all four items had to be completed. Then, participants were left alone for a few minutes to self-complete the four items. For the patients that were unable to self-complete the WELSH or did not answer the 4 items after the initial explanation, the interviewer noted the requirement for a second round of similar explanations on the file and asked to patient to complete the tool again.

## Questionnaire completion, detection of errors and score calculation

Each interviewer was asked to check the WELSH for completion and eventual errors. Errors were defined as double or missing answer at one of the four items, or paradoxical answer of

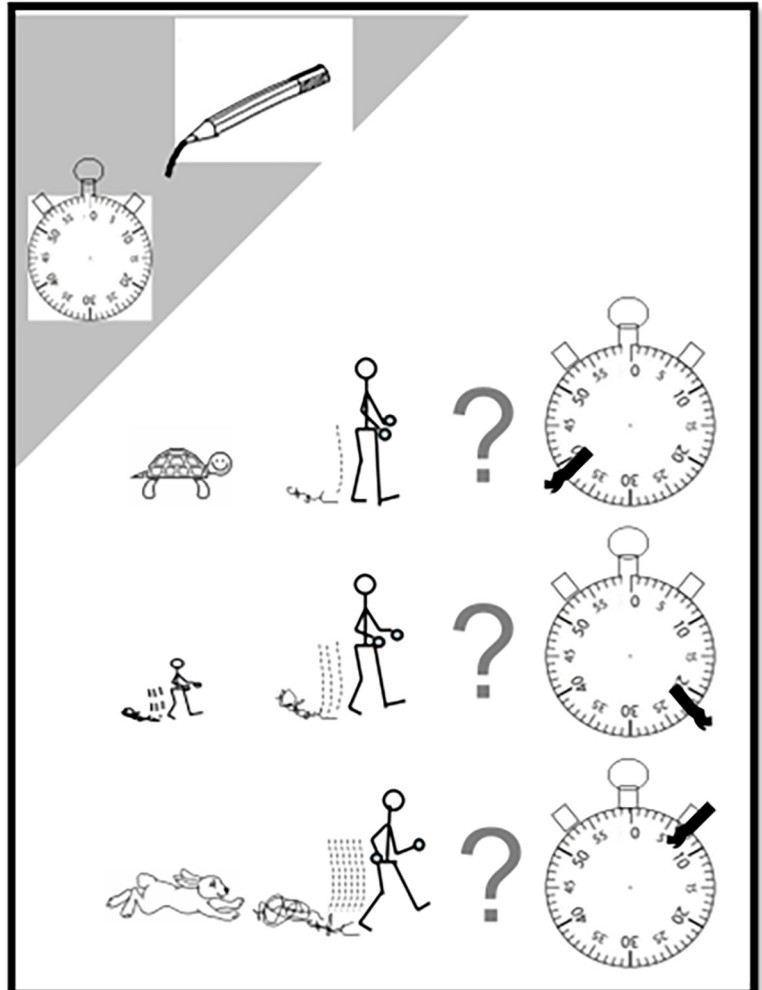

**Fig 1. The first page of a completed WELSH tool.** The first page with the first 3 items of the questionnaire. Republished from [12] under a CC BY license, with permission from Clinics Cardive Publication PTY Ltd, Copyright © 2019, Clinics Cardive Publishing: All rights reserved.

one of the first three items. An answer was considered paradoxical if the declared duration capacity was higher at a higher speed than the duration declared for the slower speed. Errors were noted in a color different from the original pencil to allow calculation of the number of scoring errors. Thereafter, interviewers had to score the filled WELSH by mental calculation and add the calculated score to the participant's sheet as shown in Fig 3.

## Six minutes' walk test

After completion of the WELSH, each participant was required to perform a 6-minute walking test. This walking test was carried out on a flat, open area around a 30-meter-long walking circuit delineated on the ground with the rope and the plastic cones. Each group of interviewers could perform the test in an area of their choice, provided that it was flat and devoid of obstacles. The participant walked back and forth around the plastic cones, and the chronometer was not stopped if patients needed to temporarily stop during the test. At 6 minutes, the interviewer stopped the test, calculated the maximal walking distance (MWD) covered by the participant in meters, and reported it on the participant's sheet.

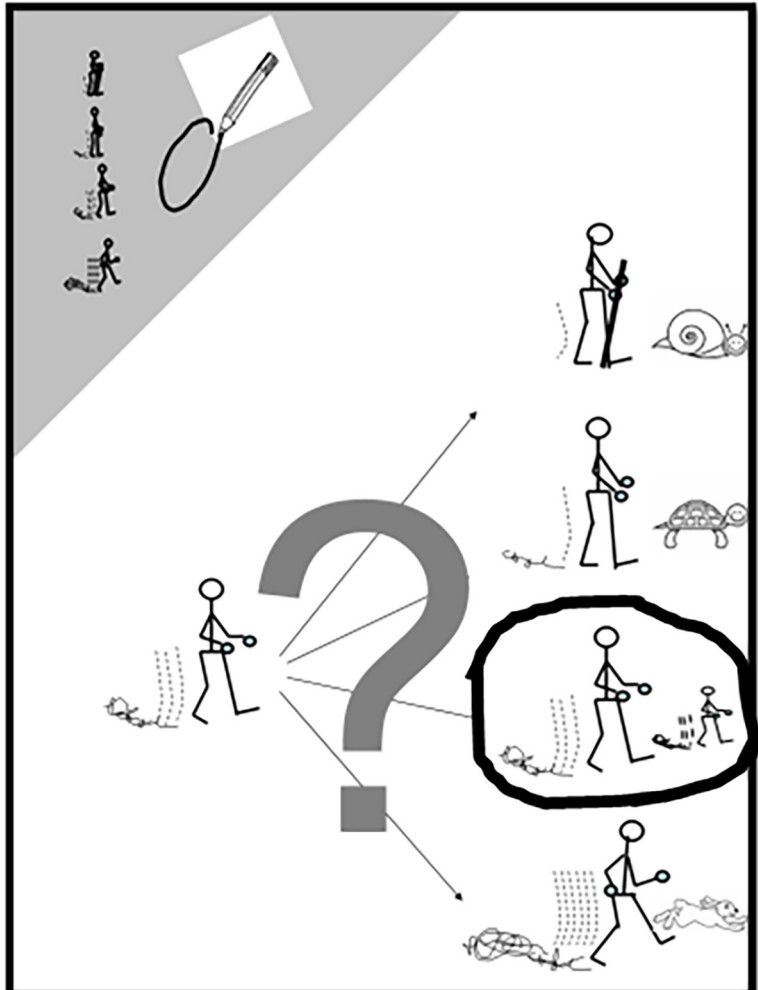

**Fig 2. The second page of a completed WELSH tool.** The second page with the fourth item of the questionnaire asking about usual walking pace. Republished from [12] under a CC BY license, with permission from Clinics Cardive Publication PTY Ltd, Copyright © 2019, Clinics Cardive Publishing: All rights reserved.

## Correction of completed files

Each of the paper records was checked again by the principal investigator for completion and eventual errors missed by the interviewers, or errors in the calculation of the WELSH score.

## Sample size and statistical analysis

Results are presented as mean ± standard deviation (SD) or as number of observations and percentages 95% confidence interval (95%CI) of the percentage are reported when appropriate. Chi-square tests were used to compare the number of filling errors in perspective of literacy. We aimed to be able to analyze data according to gender and four different literacy subgroups (8 possible subgroups), to validate our main hypothesis of a correlation of 0.40 with $\alpha$ = 5% in bilateral test and $\beta$ = 20%, at least 47 subjects per sub-group were needed. Considering possible filling errors with non-usable questionnaires, we needed 60 subjects per group. Differences between gender were tested with Chi$^2$ tests for categorial parameter and t-tests for continuous parameters. The number of correctly completed questionnaires was analyzed to

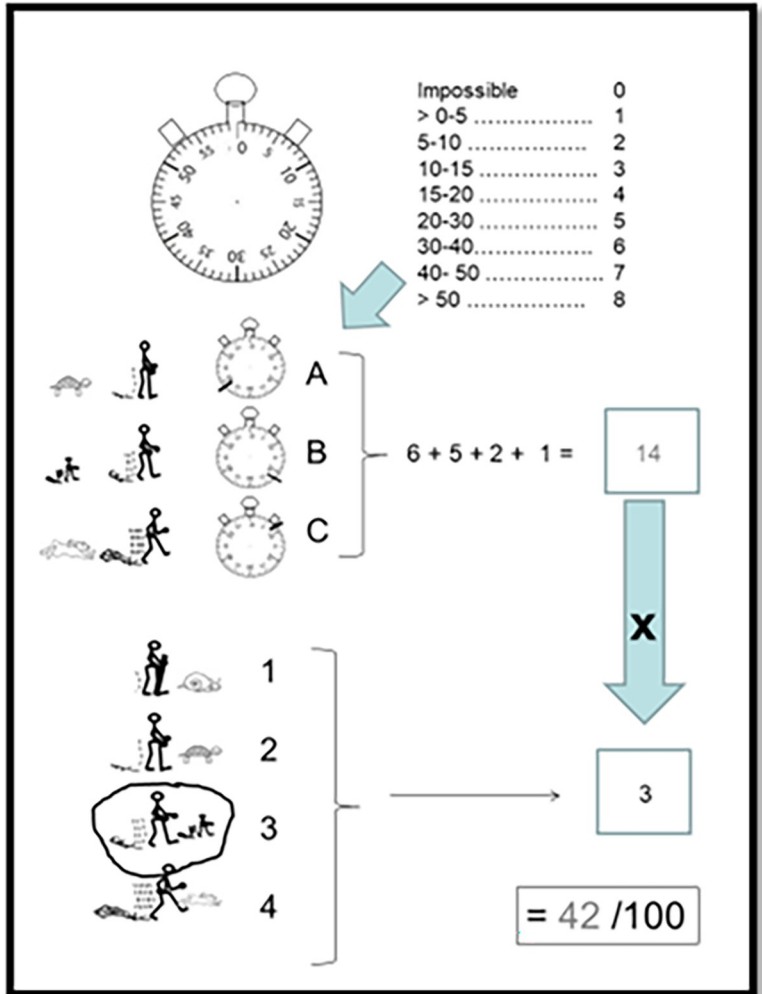

**Fig 3. An example of a questionnaire completed with the WELSH score calculation.**

assess the applicability of the WELSH questionnaire. Then, to achieve all the objectives of the study we performed a Spearman "r" correlation between the WELSH and MWD for the whole studied population and within the different subgroups, and a Fisher's Z test for their comparisons. P was adjusted for multiple testing issue using Benjamini-Hochberg procedure, which allows a control of the False Discovery Rate [13]. All statistics were performed with the SPSS V15.0; (SPSS Inc. USA).

## Results

### Population

During the month of recruitment, 1825 participants were recruited, with 1723 remaining available for the analysis (Fig 4).

### Feasibility of the WELSH score and completion errors

Among the 1723 available WELSH datasheets, 1523 (88.4%, 95%CI: 86.8%, 89.8%) were successfully self-completed by the participants after the first round of oral explanation. Of these

1523 filed WELSH, 1474 needed no correction, 42 included one error, and the others 7 included two or more errors.

Of the 200 participants that could not complete the WELSH alone, after a second explanation sixty participants completed the questionnaire alone without errors, 135 completed WELSH, participants made one (n = 94), two (n = 25), three (n = 16) errors at this second round, and five were still unable to complete the questionnaire alone or needed help for all the items.

The percentage of questionnaires with participant filling-in errors corrected by the investigator decreased with the decrease in educational level (p <0.001). It was 113 of 524 answers (21.6%, 95%CI: 18.3%, 25.3%) for participants that never went to school, 32 of 233 answers (13.7%, 95%CI: 9.9%, 18.7%) for the primary level, 40 of 433 answers (9.2%, 95%CI: 6.9%, 12.3%) for the secondary level, and 21 of 533 answers (3.9%, 95%CI: 2.6%, 6.0%) for the university level. After the second round of explanations another 140 of the 200 patients made one (n = 94) or multiple (n = 46) errors.

### Mental calculation and scoring

Of the available 1723 observations, only 1156 were scored by mental calculation by the interviewers. Only 10 of these 1156 mental calculations were wrong. Then 1146 of the 1156 (99.2%, 95%CI: 98.4%, 99.6%) were correctly scored by mental calculation.

### Correlation of WELSH to walking distances

The average WELSH score was 53 ± 22. No adverse event occurred due the 6-minute walking tests. The average MWD was 383 ±142 m. The distribution of WELSH scores and of MWD, are presented in (Fig 5).

The spearman correlation coefficient between the WELSH score and the MWD for all participants was r = 0.567 (p <0.001) as shown in (Fig 6) with MWD (m) = 3.6·WELSH score +194.

### Effect of gender and school levels

Spearman coefficients ranged from 0.291 to 0.576 for MWD (All p values <0.001) with the highest coefficients observed in illiterate male and female participants (Table 2). Correlation was particularly low in superior school females with "r" values significantly lower than superior school males (p = 0.001) and from females with no school education (p <0.001) or primary school education (p = 0.008) and secondary school (p = 0.011). Of importance is to note that the group of females that attended superior school level was the one showing the highest average MWD, dramatically reducing the range of recorded MWD values.

### Discussion

The WELSH visual tool is to date the only functional capacity assessment tool tested in a population with a high proportion of illiterate people. Its correlation to MWD is in the highest range of correlations observed between measured distances and the WIQ [1,14–16].

In all subgroups, we found significant correlation coefficients between the WELSH score and the 6-minute MWD. Although the number of errors decreased with educational level, it remained acceptable even in participants with the lowest scholar levels. Of interest is to note that previous studies focused mainly on patients with lower extremity arterial disease, while the present work was performed in the community with a large variety of morbid conditions.

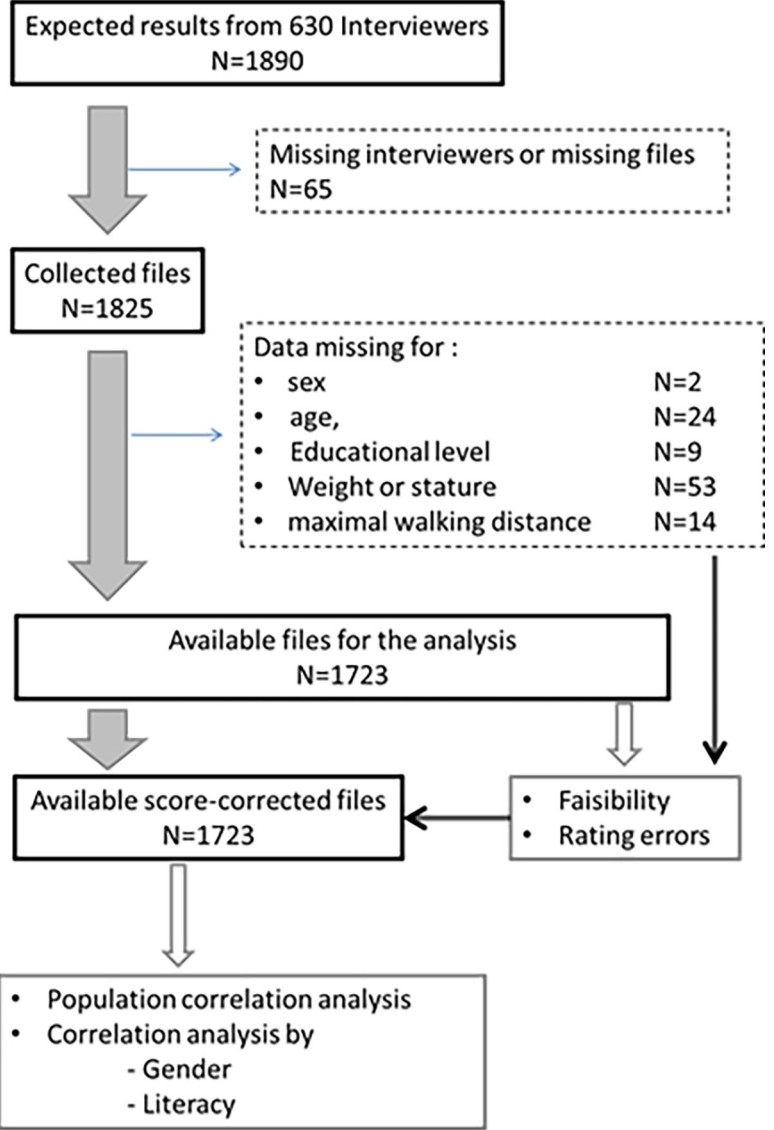

**Fig 4. Flowchart of all included participants.** The characteristics of included participants are presented in (Table 1). Note that 757 (43.9%, 95%CI: 41.6, 46.3) of these 1723 participants never went to school or attended only elementary school. Results are presented as mean ± SD or numbers of observation (%).

Nevertheless, our study has some limits. The first is related to the lack of objective proof of the morbid conditions. This is important because diseases may have different consequences on the ability to walk [17]. Nevertheless, in perspective of the healthcare organization and poor technicity in Burkina-Faso it is unlikely that we could have recorded documented evidence of diseases, even for patients recruited at healthcare facilities. The second limit is the fact that we recruited participants in a big city. Whether or not the WELSH operates similarly on a rural population, remains to be studied. Third limitation is related to the tool itself. Conceived in an empirical approach with drawings, it requires a preliminary explanation, which makes it not strictly self-administrable. Similarly, its design and scoring in an empirical approach means that it could probably be improved for greater reliability or improved correlation with objective measures of MWD. The fact that we did not account for factors such as age, weight, sex

**Table 1. Characteristics of the participants.**

| Characteristics | Total population N = 1723 | Males N = 969 | Females N = 754 | p |
|---|---|---|---|---|
| **Age (years)** | 46.5 ± 18.4 | 44.9 ± 19.1 | 48.6 ± 17.2 | 0.001 |
| **School level** | | | | |
| • **Never been to school** | 524 (30.4) | 216 (22.3) | 308 (40.6) | 0.001 |
| • **Primary** | 233 (13.5) | 125 (12.9) | 108 (14.3) | 0.391 |
| • **Secondary** | 433 (25.1) | 242 (25.0) | 191 (25.3) | 0.865 |
| • **University** | 533 (30.9) | 386 (39.8) | 147 (19.5) | 0.001 |
| **Weight (Kg)** | 70.5 ±12.7 | 70.7 ± 11.0 | 70.2 ± 14.5 | 0.426 |
| **Height (centimeter)** | 169.3 ± 8.6 | 173.0 ± 7.6 | 164.5 ± 7.2 | 0.001 |
| **Body mass index (kg/m²)** | 24.7 ± 4.5 | 23.6 ± 3.6 | 26.0 ± 5.2 | 0.001 |
| **Smokers** | 150 (8.8) | 129 (13.5) | 21 (2.8) | 0.001 |
| **Current pathology:** | | | | |
| **Hypertension** | 266 (15.4) | 103 (10.6) | 163 (21.6) | 0.001 |
| **Diabetes mellitus** | 78 (4.5) | 37 (3.8) | 41 (5.4) | 0.109 |
| **Arthrosis** | 181 (10.5) | 65 (6.7) | 116 (15.4) | 0.001 |
| **Pulmonary diseases** | 56 (3.3) | 24 (2.5) | 32 (4.2) | 0.040 |
| **Sickle cell disease** | 39 (2.3) | 19 (2.0) | 20 (2.7) | 0.338 |
| **WELSH Score** | 53 ± 22 | 58 ± 22 | 47 ± 22 | 0.001 |
| **Maximal walking distance (m)** | 383 ± 142 | 419 ± 138 | 336 ± 132 | 0.001 |

and height may clearly appear a limitation of the study. Similarly, it is likely that different time intervals and different points for each interval could improve the correlation with measured distances. Obviously, this remains to be done and might improve the results. Nevertheless, our aim was primarily to have scoring rules that are very easy to memorize and a score that is very easy to calculate by mental calculation. Then, although it might be of interest to search for other scoring rules in the future, these rules must remain very simple. Further, since the correlation is already in the high range of previously proposed questionnaires, we advocate that the gain in correlation that might result from other scoring rules will likely remain limited. Forth, the fact that speed is considered "relative to the people of the same age, family or friends" could be considered an issue with the idea that any two participants may be applying different criteria to define similar scale value. We advocate that, on the contrary, this is of major advantage to make the tool conceptually adapted (a hopefully relatively insensitive) to age. Another limitation is about the use of the 6-minutes' walk test as the only test of objective assessment of functional walking ability. The correlation of the WELSH score with other tests of objective assessments of functional walking ability, such as the treadmill test or Global Positioning System (GPS) [18], could have strengthened our results but was not adapted on a community based approach. Last the use of several evaluators results an inter-evaluator error for all assessments based on their experience, motivation, gender, age etc. . .. Indeed, personality characteristics of practitioner significantly impact their work engagement [19]. Presently, the evaluations were all performed by novice students only, but we did not record their gender or degree of motivation. Further, due to the high number of evaluators adjusting for evaluators does probably make little sense, because each observer included only 3 participants. The fact that novice students performed the tests is also the reason why, for security reasons we excluded patients with severe co-morbid conditions. Future studies under medical supervision should be performed to test the use of the WELSH as a screening tool or in various medical or surgical conditions, as well as to define which score would suggest the need for specific medical or surgical intervention. Overall, future studies are required to assess the sensibility of the WELSH to changes in clinical status.

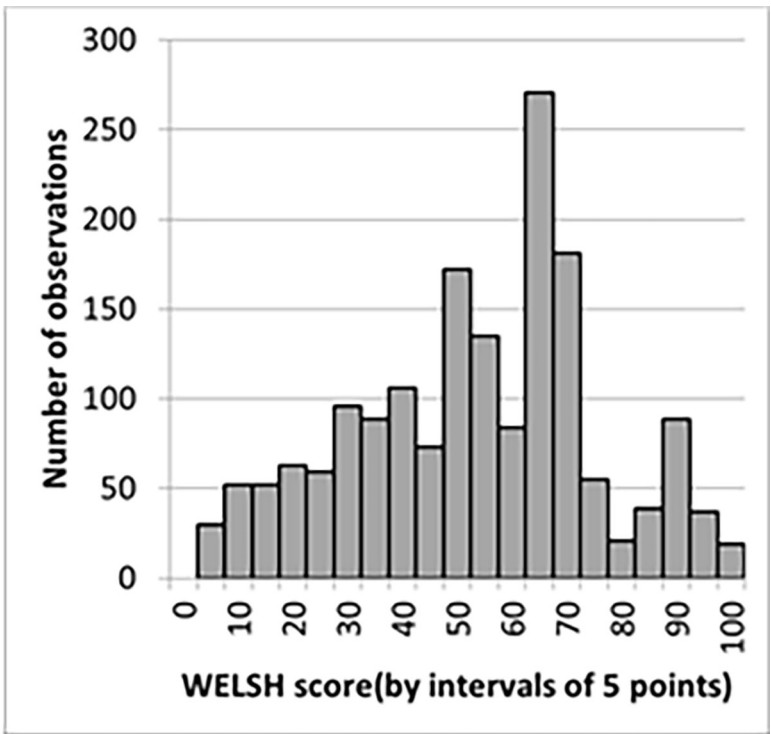

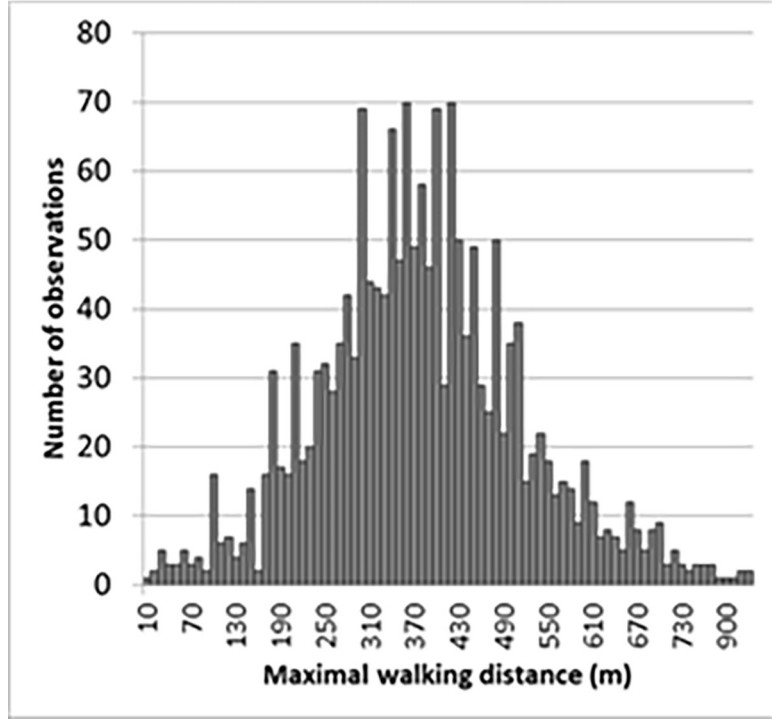

**Fig 5.** Distribution of Walking Estimated Limitation Calculated by History (WELSH) (upper panel) and of maximal walking (lower panel) in the studied population. WELSH scores are by intervals of 5 points and distances are by interval of 10 meters.

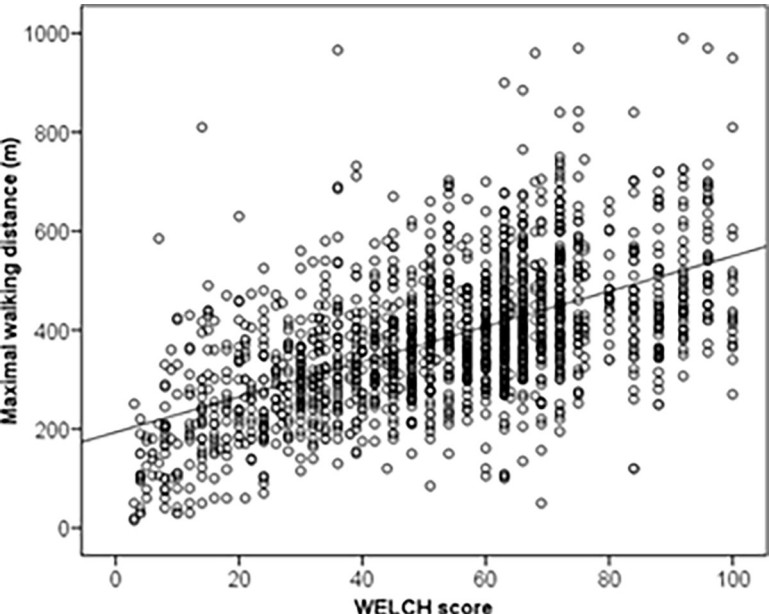

**Fig 6. Scatterplot of the maximal walking distance (lower panel) with WELSH score among the 1723 participants with linear regression line.**

**Table 2. Mean and standard deviation of maximal walking distance (MWD) and WELSH score and coefficient of correlation (r), slope and intercept of the regression linear analysis between MWD and WELSH score within each sub-group as a function of sex and level of education.**

| Sex | Level of education | Number | MWD (m) | Score | r | Slope | origin | P* |
|---|---|---|---|---|---|---|---|---|
| Females | Never been to school | 216 | 351 ± 136 | 47 ± 24 | 0.576 | 3.26 | 196 | < .001 |
| | Primary school | 125 | 376 ± 129 | 52 ± 24 | 0.520 | 2.59 | 241 | < .001 |
| | Secondary school | 242 | 428 ± 127 | 60 ± 20 | 0.469 | 3.12 | 242 | < .001 |
| | University | 386 | 466 ± 129 | 66 ± 16 | 0.291 | 2.67 | 291 | < .001 |
| Males | Never been to school | 308 | 294 ± 117 | 41± 22 | 0.429 | 3.04 | 171 | < .001 |
| | Primary school | 108 | 338 ± 147 | 46 ± 25 | 0.490 | 3.17 | 192 | < .001 |
| | Secondary school | 191 | 360 ± 120 | 50 ± 19 | 0.511 | 2.17 | 252 | < .001 |
| | University | 147 | 390 ± 137 | 55 ± 18 | 0.552 | 3.86 | 176 | < .001 |

*P was adjusted for multiple testing issue using Benjamini-Hochberg procedure.

## Conclusion

The WELSH is feasible on the community by a wide variety of non-expert interviewers. In this context, it correlates with the MWD estimated by the 6-minute walk test, even for people with little or no schooling, but given different coefficients and different levels of correlation for different subgroups the prediction equation would differ between subgroups. While we show that it might be an interesting tool for epidemiological studies in adults, whether or not it could be used in children remains to be determined.

## Supporting information

**S1 Data.**
(XLS)

## Acknowledgments

This study was conducted with the help of medical and pharmacy students from the Nazi BONI University to whom we are grateful. The authors thank Dr B Vielle and J Riou for statistical advice.

## Author Contributions

**Conceptualization:** Wendsèndaté Yves Sempore, Nafi Ouedraogo, Salifou Gandema, Pierre Abraham.

**Formal analysis:** Alassane Ilboudo.

**Investigation:** Wendsèndaté Yves Sempore, Nafi Ouedraogo.

**Methodology:** Wendsèndaté Yves Sempore, Nafi Ouedraogo, Samir Henni, Téné Marceline Yameogo, Pierre Abraham.

**Software:** Samir Henni.

**Supervision:** Salifou Gandema, Téné Marceline Yameogo, Pierre Abraham.

**Validation:** Wendsèndaté Yves Sempore, Alassane Ilboudo, Téné Marceline Yameogo, Pierre Abraham.

**Writing – original draft:** Wendsèndaté Yves Sempore, Pierre Abraham.

**Writing – review & editing:** Wendsèndaté Yves Sempore, Nafi Ouedraogo, Pierre Abraham.

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
