## [Decision Letter · Decision Letter 0]

7 Jul 2021

PONE-D-21-07947

The Walking Estimated Limitation Stated by History (WELSH) visual tool is applicable and accurate to determine walking capacity, even in people with low literacy level.

PLOS ONE

Dear Dr. SEMPORE,

Thank you for submitting your manuscript to PLOS ONE. After careful consideration, we feel that it has merit but does not fully meet PLOS ONE’s publication criteria as it currently stands. Therefore, we invite you to submit a revised version of the manuscript that addresses the points raised during the review process.

We look forward to receiving your revised manuscript.

Kind regards,

Sinan Kardeş, M.D.

Academic Editor

PLOS ONE

Journal Requirements:

Reviewers' comments:

Reviewer's Responses to Questions

**Comments to the Author**

1. Is the manuscript technically sound, and do the data support the conclusions?

Reviewer #1: Yes

Reviewer #2: No

2. Has the statistical analysis been performed appropriately and rigorously? 

Reviewer #1: Yes

Reviewer #2: No

3. Have the authors made all data underlying the findings in their manuscript fully available?

Reviewer #1: No

Reviewer #2: Yes

4. Is the manuscript presented in an intelligible fashion and written in standard English?

Reviewer #1: Yes

Reviewer #2: No

5. Review Comments to the Author

Reviewer #1: This is an interesting study describing the feasibility of WELSH questionnaire in subjects with different literacy levels. It is also reported the correlations between WELSH score and 6-minute walk test performance. The topic is remarkably interesting bringing a solution to assessment of walking capacity by questionnaire in illiterate persons, which is common in several countries.

Comments:

1) I really did not understand the questionnaire. I did not understand how the clock is used in this estimative. Please, provide sufficient details to understand the instrument.

2) In same way, what are the instructions provided to the participants. Please, describe in detail what was sad in first and second rounds.

3) The use of several evaluators includes an inter-evaluator error for all assessments. It should be described. For example, the relationships were similar when the tests were applied for novice and senior students? Men and women? Etc. Given your design, this is a main point.

4) Please provide details on how the training for these students happened. It was during class? How many students per class? Did they receive a material? Did they perform some understanding test? What instructions were done regarding the recruitment? How can we ensure that the data was really collected?

5) Same points for 6MWT. Did participants perform the test in the same corridor? What instructions were done? The first stop is not a recognized marker in the test. I suggest excluding it.

6) How anthropometric and demographic data were assessed? Please provide more detail on how information from table 1 were obtained.

7) The background to use 400m as a criterion is frail. I suggest converting the data as a percentage of predicted that adjust for sex, age and BMI (please see Ann Vasc Surg 2021 Jan;70:258-262). This adjusted data could be used for correlations.

8) A table with the data regarding the correct filling by literacy level analyzing with a chi-square test would be useful.

9) I strongly suggest removing the suggested equation from the manuscript. Neither the design nor the statistical methods were robust enough for it.

10) Figure 7 must be improved. The use of 3D strategy difficult the interpretation. The vertical axes do not have title. It is also not clear what is the difference between the left and right figures. This data could be presented in a table.

11) It is not clear why higher correlation coefficients were observed in illiterate participants. I expected at least similar results between literate and illiterate, as literate persons are also able to understand the images.

12) I suggest attenuating the statement of validity of questionnaire in the conclusion. The methods are not robust enough for it.

Reviewer #2: This paper purports to contribute data supporting the use of a visual tool that will enable individuals to self identify whether they have a walking limitation. It evaluates whether the tool is appropriate for individuals with widely varying levels of literacy. However, the methods and descriptions of the project do not not technically sound and are not well presented. Little justification is provided on how the tool should be used in clinical or research settings, and modest correlations (if the purpose is to demonstrate "Applicable and accurate walking capacity") are reported as constituting sufficient evidence.

The WELSH tool

The tool itself is described in a very confusing manner. The authors report that another paper demonstrates the validity of the approach, but there is no summary of validation data, nor a listing of how the tool can be used to benefit patients or society. The tool employs a scaling strategy that seems ad hoc and makeshift. Patients make a mark of a clock to identify the perceived maximum walking time for three different walking speeds, and then made self rating of their own pace. Oddly, if the rating was less than 20 minutes, 1 point was added for each 5 minutes of rating If it were more than 20 minutes 1 point was assigned for each 10 minutes. This seems arbitrary, and I wondered why some logarhythmic or other strategy might have provided a more comprehensible scale. The results placed each individual on an 8 point scale, then utilized as an equal interval scale and summed across the three different speeds. Then (also odd in my view) this sum was weighted (multiplied) by the self rating of walking capacity (a 4 point scale). This final score was taken to be a measure of walking capacity. (I confess that I am not familiar with previous strategies of self reported physical capacities, the the current strategy may be widely employed by clinicians. However, the statistical treatments of these unusual scale values (as presented) is not warranted.

Statistical Analysis

The paper reports P values for gender differences in Table 1 which contains a mx of categorical and continuously scaled variables. Nowhere is the statistical test described (assuming the authors used t-tests and chi-squares, but there is no presentation of the test statistics. It is impossible to tell, in many instances, the precise statistical tests that were run and what the results tell us. Also, the mean scores in some comparisons are presented with one decimal place, and no decimal places in others. It was not possible to make much sense out of sentences like, "We aimed to be able to analyze data according to gender and four different literacy subgroups (8 possible subgroups)to validate our main hypothesis of a correlation 0.40 with alpha = 5% and beta.... " This suggests that .4 is the level of correlation the authors felt necessary to demonstrate an acceptable level and the required Ns are very small compared to the total N comprising the study. (The low p-values -- all <.001-- suggests an over-powered study rather than a strong effect). The authors use P-values as representations of the strength of effects of different subgroups.

Regressions seem to be presented in the discussion of figure 6, but the statistics are not well described. There needs more discussion of the statistical approaches here. Authors should consider fitting the regression lines to the plots in Figure 6 which would provide a better visual representation of the strength of the linear relationships. Also in there conclusion to the paper, the statement, "In routine practice, the MWD in meters can be roughly estimated as 4 * the WELSH score +150 meters. Why not use the actual regression result coefficients? Also, given different levels of correlation for different subgroups suggests that this prediction equation would not be valid across subgroups.

Figure 7 seems to contain the heart of the data analysis, but its presentation is problematic. First Figure 7 is presented but not adequately described or discussed, other than the use of phrases like, "correlation was particularly low (P=0.025)".

In Figure 7, there are no labels for the Y axes and no labeling indicating that the panels on the right are of percentages of participants who were able to walk more than 400 meters during the 6-minute walking test (used to validate the too, nor that the panels on left show correlation coefficients.

Finally, Figure 7 employs an unacceptable 3-D graphing strategy. It is very difficult to determine the nature of the group comparisons.

In sum, while this paper asks an interesting question regarding the availability of a tool appropriate for individuals with limited reading ability, its statistical treatment of the data is quite weak, the scaling strategy for the WELSH needs greater justification, and a better discussion of the legitimate uses of the tool, given the findings, and the limits should be provided. As well, the paper is difficult to follow at times, and the statistics are not presented in conventional formatting.

6. PLOS authors have the option to publish the peer review history of their article (what does this mean?). If published, this will include your full peer review and any attached files.

Reviewer #1: No

Reviewer #2: No

---

## [Author Response · Author response to Decision Letter 0]

30 Sep 2021

Answers to reviewer #1: This is an interesting study describing the feasibility of WELSH questionnaire in subjects with different literacy levels. It is also reported the correlations between WELSH score and 6-minute walk test performance. The topic is remarkably interesting bringing a solution to assessment of walking capacity by questionnaire in illiterate persons, which is common in several countries.

Thank you for this excellent summary of our work.

Comments:

1) I really did not understand the questionnaire. I did not understand how the clock is used in this estimative. Please, provide sufficient details to understand the instrument.

We thank the reviewer for this comment and have tried to complete the revised version according to this suggestion. We apologize that the first version of the manuscript was probably not providing sufficient details to allow an understanding of the WELSH concept. We had the questionnaire already published but do understand that a better explanation was required to help the readers. The participants just had to put a mark on the required duration on the clock. We hope that the manuscript is now easier to read and are ready to complete the text further if necessary.

2) In same way, what are the instructions provided to the participants. Please, describe in detail what was sad in first and second rounds.

We thank the reviewer for this comment and have tried to complete the revised version according to this suggestion. 

3) The use of several evaluators includes an inter-evaluator error for all assessments. It should be described. For example, the relationships were similar when the tests were applied for novice and senior students? Men and women? Etc. Given your design, this is a main point.

Thank you for this comment. The evaluations were all performed by novice students only and it was indeed one of the interests of the present study that was probably not sufficiently described or explained. We must recognize that we have not recorded the fact that the students were males or females. A paragraph and a reference have been added to the text to underline this important point. The question is clearly of interest and future studies might be done for that. Thankfully, due to the high number of evaluators adjusting for evaluators does probably make little sense (because each observer included only 3 participants) but it is a clever advice for our future protocols and if a limited number of observers include a lot of participants adjustment for observer characteristics should be needed. This has been added as an important point of discussion to the manuscript.

4) Please provide details on how the training for these students happened. It was during class? How many students per class? Did they receive a material? Did they perform some understanding test? What instructions were done regarding the recruitment? How can we ensure that the data was really collected?

Thank you for this important comment and again sorry that the initial manuscript was quite poor on this description. The latest point is of interest and explains why the student were working by groups of two to be sure that data were really collected.

5) Same points for 6MWT. Did participants perform the test in the same corridor? What instructions were done? The first stop is not a recognized marker in the test. I suggest excluding it.

Sorry that this point was unclear. No clearly not, the students did not do the tests in the same corridor… this is why they were provided a 30 m rope with cones to each group. The distance to first stop is used in the vascular literature with 6 -min walk tests but we do agree that it is not routinely used. Further removing it from the manuscript simplifies the manuscript and we thank the reviewer for this suggestion.

6) How anthropometric and demographic data were assessed? Please provide more detail on how information from table 1 were obtained.

We humbly ask the reviewer to consider the context of the study that was performed in one of the poorest countries in west Africa. It was clearly impossible to have scales for weight the anthropometric values reported are in most cases those self-reported by the patients. Note that for stature it is noted on each ID card and was retrieved from the ID. This is a real-life situation in a very low-income country. We did our best to perform the study on rigorous ethical standard and methodology, but we also had to be pragmatic on feasibility. Should this point be added to the discussion? We did not do it because we did not want to provide a miserabilist view of this work because we do think that it is a unique study, despite obvious contextual limitations.

7) The background to use 400m as a criterion is frail. I suggest converting the data as a percentage of predicted that adjust for sex, age and BMI (please see Ann Vasc Surg 2021 Jan;70:258-262). This adjusted data could be used for correlations.

Thank you for this comment used for PAD We choose 400 m because a large series of recent studies proposed the 400-m walking test as a way of defining walking limitation PMID: 34283660 PMID: 34283660, PMID: 33066134, PMID: 34283660. Further there are various equations proposed in the literature to estimate walking distance from Age/sex BMI, or age height weight and sex. When comparing the observed results of our group to these equations we found a mean difference with the expected results of -217 m -172m and -360m for three of the selected equation respectively (The one from Brazil being the first result). Clearly our population has lower results than expected from a general population which is logical due to the recommendation done to recruit. The second question that arises there is: what should be considered the normal or abnormal value? Indeed only few of the available equation provide their standard deviation from the mean and when they do the unsolved question is “should we consider a low value be minus 1SD or minus 1.5SD or minus 2SD. For the reviewer information the figures below represent the differences observed between measured distances and theoretical equations for 3 equations proposed in the literature (among which the one that you proposed is the first one), the other two are:

• Troosters T, Gosselink R, Decramer M. Six minute walking distance in healthy elderly subjects. Eur Respir J [Internet]. 1999 Aug;14(2):270–4. Available from: http://erj.ersjournals.com/content/14/2/270

• Nusdwinuringtyas N, Widjajalaksmi, Yunus F, Alwi I. Reference equation for prediction of a total distance during six-minute walk test using Indonesian anthropometrics. Acta Med Indones [Internet]. 2014 Apr;46(2):90–6. 

 As shown the average distance that was obtained was lower than the average estimated MWD but clearly depended on which equation was used. Following the reviewer suggestion, we have removed reference to 400 m as the distance to discriminate the presence from the absence of limitation.

We propose below just for the reviewer information an analysis based on the correlation of the WELCH score against the difference from theoretical value based on the three equations that were used for the above analysis.

Results were r=0.402 (p=0.001) with “difference from theoretical MWD” = 2.72*WELCH-337

Results were r=0.576 (p=0.001) with “difference from theoretical MWD” = 3.97*WELCH-383

Results were r=0.170 (p=0.001) with “difference from theoretical MWD” = 1.02*WELCH-414

None of these do better that the use of MWD not adjusted for theoretical values and the problem there would be to justify the used of one of the models rather than another.

8) A table with the data regarding the correct filling by literacy level analyzing with a chi-square test would be useful.

The data are provided page 9 line 180-194. Since we added a table to the manuscript we have left the results as a text.

9) I strongly suggest removing the suggested equation from the manuscript. Neither the design nor the statistical methods were robust enough for it.

We have removed this from the conclusion as suggested

10) Figure 7 must be improved. The use of 3D strategy difficult the interpretation. The vertical axes do not have title. It is also not clear what is the difference between the left and right figures. This data could be presented in a table.

We have changed table 7 to account for the present suggestion.

11) It is not clear why higher correlation coefficients were observed in illiterate participants. I expected at least similar results between literate and illiterate, as literate persons are also able to understand the images.

We completely agree with the reviewer and have no satisfactory explanation for this. Could it be that subjects with high level of literacy were less careful about the filling considering that this was a too simple tool for their level of education? Could it be that the range of observed value was lower in educated patients (Which almost automatically trends to reduce the quality of the correlation)? Could it only be by chance?

12) I suggest attenuating the statement of validity of questionnaire in the conclusion. The methods are not robust enough for it.

We have smoothed our conclusion to account for this suggestion.

 

Answers to reviewer #2: This paper purports to contribute data supporting the use of a visual tool that will enable individuals to self identify whether they have a walking limitation. It evaluates whether the tool is appropriate for individuals with widely varying levels of literacy. However, the methods and descriptions of the project do not not technically sound and are not well presented. Little justification is provided on how the tool should be used in clinical or research settings, and modest correlations (if the purpose is to demonstrate "Applicable and accurate walking capacity") are reported as constituting sufficient evidence.

Thank you for this summary of our work. We agree that the present study is only one step forward in the validation of the WELSH tool, that will need external validation, analysis of sensitivity to changes of reliability of internal consistency, Evaluation of the presence of potential ceiling effect, etc… Concerning the modest correlation we wish to underline that the present tool is clearly in the same range as most of other available tools (including the WIQ). We apologize that the description of how the tool was used in the present study (and could be used by others) was missing. This has also been underlined by reviewer 1 and we have completed the manuscript to account for these perfectly justified criticisms.

The WELSH tool

The tool itself is described in a very confusing manner. The authors report that another paper demonstrates the validity of the approach, but there is no summary of validation data, nor a listing of how the tool can be used to benefit patients or society. 

This comes down to the previous comment and the manuscript has been completed to provide more information on this important point.

The tool employs a scaling strategy that seems ad hoc and makeshift. Patients make a mark of a clock to identify the perceived maximum walking time for three different walking speeds, and then made self-rating of their own pace. Oddly, if the rating was less than 20 minutes, 1 point was added for each 5 minutes of rating If it were more than 20 minutes 1 point was assigned for each 10 minutes. This seems arbitrary, and I wondered why some logarythmic or other strategy might have provided a more comprehensible scale. The results placed each individual on an 8-point scale, then utilized as an equal interval scale and summed across the three different speeds. Then (also odd in my view) this sum was weighted (multiplied) by the self-rating of walking capacity (a 4 points scale). This final score was taken to be a measure of walking capacity. (I confess that I am not familiar with previous strategies of self-reported physical capacities, then the current strategy may be widely employed by clinicians. However, the statistical treatments of these unusual scale values (as presented) is not warranted.

We thank the reviewer for this comment that requires some extensive explanation. Initially the first tool that was developed was the EACH-Q questionnaire that was built on the concept that patients have reduced capacity when walking faster and that patients may apparently report satisfactory walking capacity if their usual walking speed is slow. Later, the EACH-Q has been simplified and slightly changed to the WELCH questionnaire with the idea of making a toll that would result in a score ranging 0 (inability to walk) to 100 (no limitation). The durations proposed for each walking pace followed an exponential increase. The sum of durations (scored 0 to 7) resulted in values ranging 0 to 21 and we proposed 5 different usual walking speeds… then we arbitrarily subtracted one to the sum before multiplying by 1 to 5.

In the process of transferring the concept from the textual WELCH into a visual WELSH, we faced a series of issues. First, it was clearly not easy to propose a logarithmic scale on the watch and we arbitrarily defined simple intervals that could easily be memorized by the interviewers but kept the concept that the highest duration (= or > 20 min) would represent intervals larger than short durations. Second it was relatively uneasy to represent 5 different usual walking speed and then we reduced the proposed answer to usual paces to 4 possibilities…. As a result, as the goal was to have a final score ranging 0 to 100 we had to have the first three question to have individual score that allowed a sum of 25. All these assumptions were totally arbitrary although defined on physiological concepts and it is more than likely that adjusting the coefficients proposed for the various time answers would improve the correlation with measured distance. This is indeed something that could be done from the recorded data, with for example a multilinear regression analysis. The question here is the routine use of the tool. Is gaining a few units of correlation of WELSH with 6MWT distance by applying coefficients to each answer (e.g; 2.5*the duration at low speed + 5.3 *the duration found at medium speed + 6.24 * the duration at high speed), or defining intervals of time that might represent a semi-logarithmic increase (e.g.: 2 minutes, 5 minutes, 11 minutes, 23 minutes etc… ) worth it? It would clearly result in less facility to use the tool in a routine. We completely agree that it might be of interest to confront the results of the completely arbitrarily defined score to mathematically defined coefficients. Our purpose was not there. What we only wanted to do is test the large-scale applicability to use the WELSH (define as a very simple tool both to fill and to score) and the ability of the WELSH to provide reasonable results in illiterate patients. On the scoring point of view, the fact that less than 1% of the scores calculated by student were wrong is an essential point. We doubt that a comparable result could be reached with more complex scoring rules (based on adjusted coefficients or adjusted time intervals). Please consider that the WELSH aims to be a pragmatic easy tool conceived for a pragmatic application in low-income country context with low literacy level. Clearly a more statistical approach would probably lead to improved correlation but would to our opinion lead to a decreased applicability while a more clinical and pragmatic approach is indeed very likely decreasing the quality of the correlation.

Statistical Analysis

The paper reports P values for gender differences in Table 1 which contains a mx of categorical and continuously scaled variables. Nowhere is the statistical test described (assuming the authors used t-tests and chi-squares, but there is no presentation of the test statistics. It is impossible to tell, in many instances, the precise statistical tests that were run and what the results tell us. Also, the mean scores in some comparisons are presented with one decimal place, and no decimal places in others. It was not possible to make much sense out of sentences like, "We aimed to be able to analyze data according to gender and four different literacy subgroups (8 possible subgroups) to validate our main hypothesis of a correlation 0.40 with alpha = 5% and beta.... " This suggests that .4 is the level of correlation the authors felt necessary to demonstrate an acceptable level and the required Ns are very small compared to the total N comprising the study. (The low p-values -- all <.001-- suggests an over-powered study rather than a strong effect). The authors use P-values as representations of the strength of effects of different subgroups.

On the one hand, we apologize that the statistical tests were not described and confirm that the assumption made by the reviewer are correct (Use of Chi² and t tests). On the other hand, it may have been unclear that what we dreamed of were coefficients of 0.5 to 0.6 (which is the range of coefficients observed between self-reported capacity tools and various methods of objective measurements) but we also wanted to be able to conclude that even coefficients of at least 0.40 would be statistically significant. Thereby, the calculation was based on the worth possible scenario (i.e. the worth coefficient in the smallest sub-group). Since the recruitment was unpredictable (we did not define the subjects that had to be recruited by each interviewer), The total number of subjects had to be high. This was also an interesting point to confirm that the tool was easy to use and score. Last, Yes a coefficient of 0.40 is already an acceptable level of correlation in perspective of previous results of the literature.

Regressions seem to be presented in the discussion of figure 6, but the statistics are not well described. There needs more discussion of the statistical approaches here. Authors should consider fitting the regression lines to the plots in Figure 6 which would provide a better visual representation of the strength of the linear relationships. Also in there conclusion to the paper, the statement, "In routine practice, the MWD in meters can be roughly estimated as 4 * the WELSH score +150 meters. Why not use the actual regression result coefficients? Also, given different levels of correlation for different subgroups suggests that this prediction equation would not be valid across subgroups.

Following the reviewer comment a regression line has been added to the plot. Please consider that the upper figure has been removed following reviewer one comments. As suggested by the reviewer the equation has been removed.

Figure 7 seems to contain the heart of the data analysis, but its presentation is problematic. First Figure 7 is presented but not adequately described or discussed, other than the use of phrases like, "correlation was particularly low (P=0.025)". In Figure 7, there are no labels for the Y axes and no labeling indicating that the panels on the right are of percentages of participants who were able to walk more than 400 meters during the 6-minute walking test (used to validate the too, nor that the panels on left show correlation coefficients.

Finally, Figure 7 employs an unacceptable 3-D graphing strategy. It is very difficult to determine the nature of the group comparisons.

We apologize for the poor quality of figure 7 that has been removed and replaced by a table (Table 2).

In sum, while this paper asks an interesting question regarding the availability of a tool appropriate for individuals with limited reading ability, its statistical treatment of the data is quite weak, the scaling strategy for the WELSH needs greater justification, and a better discussion of the legitimate uses of the tool, given the findings, and the limits should be provided. As well, the paper is difficult to follow at times, and the statistics are not presented in conventional formatting.

We hope that the answers provided to the different suggestions that were done to improve the initially submitted manuscript fulfill the reviewer expectation and are ready to further work on thee manuscript if some point require other clarifications.

---

## [Decision Letter · Decision Letter 1]

20 Oct 2021

PONE-D-21-07947R1The Walking Estimated Limitation Stated by History (WELSH) visual tool is applicable and accurate to determine walking capacity, even in people with low literacy level.PLOS ONE

Dear Dr. Abraham,

Thank you for submitting your manuscript to PLOS ONE. After careful consideration, we feel that it has merit but does not fully meet PLOS ONE’s publication criteria as it currently stands. Therefore, we invite you to submit a revised version of the manuscript that addresses the points raised during the review process.

We look forward to receiving your revised manuscript.

Kind regards,

Sinan Kardeş, M.D.

Academic Editor

PLOS ONE

Reviewers' comments:

Reviewer's Responses to Questions

**Comments to the Author**

1. If the authors have adequately addressed your comments raised in a previous round of review and you feel that this manuscript is now acceptable for publication, you may indicate that here to bypass the “Comments to the Author” section, enter your conflict of interest statement in the “Confidential to Editor” section, and submit your "Accept" recommendation.

Reviewer #1: (No Response)

Reviewer #2: (No Response)

2. Is the manuscript technically sound, and do the data support the conclusions?

Reviewer #1: Yes

Reviewer #2: Partly

3. Has the statistical analysis been performed appropriately and rigorously? 

Reviewer #1: Yes

Reviewer #2: No

4. Have the authors made all data underlying the findings in their manuscript fully available?

Reviewer #1: Yes

Reviewer #2: Yes

5. Is the manuscript presented in an intelligible fashion and written in standard English?

Reviewer #1: (No Response)

Reviewer #2: Yes

6. Review Comments to the Author

Reviewer #1: Dear authors, thank for your reply. Most of my concerns were addressed, but I still uncomfortable with few points.

Thank you for this excellent summary of our work.

Comments:

1) Sorry to insist, but I still not understanding the instrument. “In brief, for the first three items, the maximum walking time that can be performed for each of 3 different walking speeds (illustrated by a turtle a human and a rabbit) must be reported. Walking speeds are considered relative to the people of the same age, family or friends”. Ok, but for what distance?

2) “We choose 400 m because a large series of recent studies proposed the 400-m walking test as a way of defining walking limitation PMID: 34283660 PMID: 34283660, PMID: 33066134, PMID: 34283660”. Note that out of 4 studies, three are exactly the same (Effect of whole-body resistance training at different load intensities on circulating inflammatory biomarkers, body fat, muscular strength, and physical performance in postmenopausal women) and the other one is in cancer patients (Association between Sarcopenia and Physical Function among Preoperative Lung Cancer Patients).

3) “None of these do better that the use of MWD not adjusted for theoretical values and the problem there would be to justify the used of one of the models rather than another.” The point is not what correlates better, but that 6MWT performance is influenced by clinical and demographic parameters. It is well stablished that women, elderly, obese and smaller subjects present lower values than men, taller, normal weight and young subjects. Therefore, 400m for an elderly woman indicates a better health condition than a 400m in a young men. The use of equation is an adjustment for these factors, which is stronger than the simple correlation. The suggestion is to use the equation in which the population profile is closer of your population (age, height, obesity prevalence, etc). A threshold of 84% is often employed, however, for your linear analysis it is not needed. I think this is an additional and not a substitutive data.

Reviewer #2: This paper is dramatically improved. The authors have addressed most of my original comments. I commend the authors for the completeness with which they have addressed the reviewer comments.

1. I still have some concerns about the validity of the tool. To what purposes is the tool valid. Is it useful a a screening device? Do the scores suggest specific medical interventions? Although they do not use the term "diagnostic",there is an implied conclusion that the test has diagnostic value. A description of the situations in which the scale wold be useful wold be helpful.

2. Why eliminate participants with known walking disabilities? Would it be advantageous to show that the WELSH scores on disabled individuals to demonstrate its validity in potential disability in others?

3. Should the picture version of the test been validated with the text version of the questionnaire administered to participants who can read? This wold lend confidence that the picture version is measuring the same construct as the text version.

4. What are the inter-correlations among the four items of the test, and What are the correlation of these items with the walking score singly? Does the proposed scaling result in a higher correlation than, for example, simply using question 4?

5. I would have used a non-parametric correlation, such as Spearman, to avoid the criticism regarding the equal-interval assumptions required for a Pearson r and linear regression.

6. Finally, I question the use of a self rating as this in epidemiological studies that are listed as potential uses of the scale, where any two participants may be applying different criteria to define similar scale value, leading to epidemiological findings that wold be difficult to interpret.

7. PLOS authors have the option to publish the peer review history of their article (what does this mean?). If published, this will include your full peer review and any attached files.

Reviewer #1: No

Reviewer #2: No

---

## [Author Response · Author response to Decision Letter 1]

27 Oct 2021

Reviewer #1: Dear authors, thank for your reply. Most of my concerns were addressed, but I still uncomfortable with few points.

Comments:

1) Sorry to insist, but I still not understanding the instrument. “In brief, for the first three items, the maximum walking time that can be performed for each of 3 different walking speeds (illustrated by a turtle a human and a rabbit) must be reported. Walking speeds are considered relative to the people of the same age, family or friends”. Ok, but for what distance?

We are very uncomfortable with this comment because we are not sure that we understood it.

If the comment relies on the fact that we should have better asked for distance rather than time, we have already explained in the present manuscript (as we did before for the papers dealing with the WELCH) that evaluating time is easier than distance (specifically in open spaces)

If the comment relies on the fact that time for a defined speed may not relate to distance: mathematically distance is the product of time per speed. Then a linear relationship exists between the two for a defined speed whatever the speed might be.

If the comment relies to the fact that speed is to be understood relative to people of the same age (and might be slightly different for old or young people) it is perfectly on purpose to make the WELSH hopefully independent of age as was the WELCH (which is a text device based on the same concept but not the copy of the WELSH). A short advice has been added to the introduction to underline this point: “an adapted version of the WELCH (and not a translation of the WELCH into images)”

2) “We choose 400 m because a large series of recent studies proposed the 400-m walking test as a way of defining walking limitation PMID: 34283660 PMID: 34283660, PMID: 33066134, PMID: 34283660”. Note that out of 4 studies, three are exactly the same (Effect of whole-body resistance training at different load intensities on circulating inflammatory biomarkers, body fat, muscular strength, and physical performance in postmenopausal women) and the other one is in cancer patients (Association between Sarcopenia and Physical Function among Preoperative Lung Cancer Patients).

We apologize for the errors with the three same references. Our aim was only to underline that many authors have used the 400m limit in sarcopenia, cancer, cardiac, pulmonary epidemiology studies and suggested its use in review papers. (PMID: 20166006; PMID: 32068846; PMID: 16545950; PMID: 31987880; PMID: 27174883; PMID: 30312372, PMID: 31742368, and we could have added a lot of other references….). Apparently, we did not sufficiently underline that we had removed this 400m limit from our manuscript to avoid criticisms from people that may use the 400 m limit or that would rightly underline the issue of normalization (see comments and answers to the next point). We assume that the reviewer read the comments and answers but not the revised manuscript either in redline or clean version.

3) “None of these do better that the use of MWD not adjusted for theoretical values and the problem there would be to justify the used of one of the models rather than another.” The point is not what correlates better, but that 6MWT performance is influenced by clinical and demographic parameters. It is well stablished that women, elderly, obese and smaller subjects present lower values than men, taller, normal weight and young subjects. Therefore, 400m for an elderly woman indicates a better health condition than a 400m in a young men. The use of equation is an adjustment for these factors, which is stronger than the simple correlation. The suggestion is to use the equation in which the population profile is closer of your population (age, height, obesity prevalence, etc). A threshold of 84% is often employed, however, for your linear analysis it is not needed. I think this is an additional and not a substitutive data.

We agree to the suggestions of the reviewer about the difference that 400 m may represent pending on age, sex, etc… but we are a bit in trouble with these two comments of the reviewer because both rely on the initial version of our manuscript and not to the revision that was submitted. Indeed, all references to the 400 m have been removed from our manuscript and this point no longer applies to the revised manuscript as explained above. 

Reviewer #2: This paper is dramatically improved. The authors have addressed most of my original comments. I commend the authors for the completeness with which they have addressed the reviewer comments.

1. I still have some concerns about the validity of the tool. To what purposes is the tool valid. Is it useful a a screening device? Do the scores suggest specific medical interventions? Although they do not use the term "diagnostic",there is an implied conclusion that the test has diagnostic value. A description of the situations in which the scale wold be useful wold be helpful.

We thank the reviewer for this question that requires diverse answers for the various points. 

• The issue of purposes is very important and clearly will require future studies that were not scheduled until we had some idea of the questionnaire applicability in low literacy people. Use of the WELSH in the future will require various validations in various application domains. We are currently working on severe anemia in pregnancy in an African population as well as a project is ongoing in vascular patients. The WELSH is not licensed and future researchers are free to use and validate the tool in various populations and/or various diseases states 

• The issue of screening device is one of the possible uses of a tool applicable to a population of low literacy. 

• This issue of which score shall require specific medical interventions and a very important question although it is clearly impossible to answer to this question to date. This limit for intervention clearly depends on which disease is studied and in which population.

• That fact that we do not use the term "diagnostic", is clearly on purpose because to date we cannot claim that it has diagnostic value… at least to date. Further the potential application could also be for follow-up rather than diagnostic

• A description of the situations in which the scale would be useful would be helpful. This sentence is somehow a summary of the previous comments and we have added a short paragraph to the manuscript to account for these comments just before the conclusion

 “The fact that novice students performed the tests is also the reason why, for security reasons we excluded patients with severe co-morbid conditions. Future studies under medical supervision should be performed to test the use of the WELSH as a screening tool or in various medical or surgical conditions, as well as to define which score would suggest the need for specific medical or surgical intervention.”

2. Why eliminate participants with known walking disabilities? Would it be advantageous to show that the WELSH scores on disabled individuals to demonstrate its validity in potential disability in others?

We totally agree with the reviewer that including disabled people would have increased the range of available data, but for security reasons of non-supervised out-of-the-lab tests we wanted to have no severely disabled patients. It is clear that including disabled subjects would have enlarged the range of available data which generally trends to improve correlation coefficients. Then the coefficients obtained are possibly improved in the future. Then it clearly is of interest in the future to do studies including disabled but we could not do it here. This limit has been added to the final sentence of the discussion reported above “The fact that novice students performed the tests is also the reason why, for security reasons we excluded patients with severe co-morbid conditions”

3. Should the picture version of the test been validated with the text version of the questionnaire administered to participants who can read? This would lend confidence that the picture version is measuring the same construct as the text version.

Thank you for this question. We apologize if the manuscript was a bit confusing but the WELSH is not a picture version of the WELCH. The WELSH is not aiming at copying the text version (WELCH) but it is only based on similar concepts (1/ estimation of time instead of distance, 2/ estimation of time for different speeds, 3/: estimation of usual speed). Consistently the calculation of score is not the same. Then there is no real objective reason to compare WELSH to the WELCH rather than to the WIQ or Vascu-QOL or any of the other available text questionnaires. We have added the following advice to the introduction to avoid confusion :” (and not a translation of the WELCH into images)”

4. What are the inter-correlations among the four items of the test, and What are the correlation of these items with the walking score singly? Does the proposed scaling result in a higher correlation than, for example, simply using question 4?

We thank the reviewer for this very interesting question that was worth a try and that we did not previously do. For the reviewer suggestion the analysis based on available data shows that individual questions do not better correlate with MWD than the score. We provide the results for the reviewer information but have not included them in the text: Correlation to MWD for 1/ Time at a low speed r= 0.458; 2/ Time at average speed r=0.482; 3/ Time at a high speed r = 0.479; 4/ Usual speed r=0.488; while the correlation with the final score was r=0.567. Nevertheless, these results have not been added to the manuscript but should be added if the reviewer thinks that they are essential to the readers.

5. I would have used a non-parametric correlation, such as Spearman, to avoid the criticism regarding the equal-interval assumptions required for a Pearson r and linear regression.

Thank you for your comment. We have therefore followed your advice for all the analyses and modified these elements in the manuscript.

6. Finally, I question the use of a self rating as this in epidemiological studies that are listed as potential uses of the scale, where any two participants may be applying different criteria to define similar scale value, leading to epidemiological findings that would be difficult to interpret.

We assume that this comes down to the question of estimating for walking speeds “relative to family friends. Contrary to being a limit, we advocate that this self-rating relative to people of the same age family and friends is a major advantage to have the tool conceptually adapted to (and hopefully independent of) age. To account for this comment, the following paragraph has been added to the manuscript “Forth, the fact that speed is considered “relative to the people of the same age, family or friends” could be considered an issue with the idea that any two participants may be applying different criteria to define similar scale value. We advocate that, on the contrary, this is of major advantage to make the tool conceptually adapted (a hopefully relatively insensitive) to age.”

---

## [Decision Letter · Decision Letter 2]

12 Nov 2021

PONE-D-21-07947R2The Walking Estimated Limitation Stated by History (WELSH) visual tool is applicable and accurate to determine walking capacity, even in people with low literacy level.PLOS ONE

Dear Dr. Abraham,

Thank you for submitting your manuscript to PLOS ONE. After careful consideration, we feel that it has merit but does not fully meet PLOS ONE’s publication criteria as it currently stands. Therefore, we invite you to submit a revised version of the manuscript that addresses the points raised during the review process. Please submit your revised manuscript by Dec 27 2021 11:59PM. If you will need more time than this to complete your revisions, please reply to this message or contact the journal office at plosone@plos.org. Please include the following items when submitting your revised manuscript:A rebuttal letter that responds to each point raised by the academic editor and reviewer(s). You should upload this letter as a separate file labeled 'Response to Reviewers'.A marked-up copy of your manuscript that highlights changes made to the original version. You should upload this as a separate file labeled 'Revised Manuscript with Track Changes'.An unmarked version of your revised paper without tracked changes. You should upload this as a separate file labeled 'Manuscript'.If applicable, we recommend that you deposit your laboratory protocols in protocols.io to enhance the reproducibility of your results. Protocols.io assigns your protocol its own identifier (DOI) so that it can be cited independently in the future. For instructions see: https://journals.plos.org/plosone/s/submission-guidelines#loc-laboratory-protocols. Additionally, PLOS ONE offers an option for publishing peer-reviewed Lab Protocol articles, which describe protocols hosted on protocols.io. Read more information on sharing protocols at https://plos.org/protocols?utm_medium=editorial-email&utm_source=authorletters&utm_campaign=protocols.

We look forward to receiving your revised manuscript.

Kind regards,

Sinan Kardeş, M.D.

Academic Editor

PLOS ONE

Journal Requirements:

Reviewers' comments:

Reviewer's Responses to Questions

**Comments to the Author**

1. If the authors have adequately addressed your comments raised in a previous round of review and you feel that this manuscript is now acceptable for publication, you may indicate that here to bypass the “Comments to the Author” section, enter your conflict of interest statement in the “Confidential to Editor” section, and submit your "Accept" recommendation.

Reviewer #1: All comments have been addressed

Reviewer #2: All comments have been addressed

2. Is the manuscript technically sound, and do the data support the conclusions?

Reviewer #1: Yes

Reviewer #2: Yes

3. Has the statistical analysis been performed appropriately and rigorously? 

Reviewer #1: Yes

Reviewer #2: Yes

4. Have the authors made all data underlying the findings in their manuscript fully available?

Reviewer #1: (No Response)

Reviewer #2: Yes

5. Is the manuscript presented in an intelligible fashion and written in standard English?

Reviewer #1: (No Response)

Reviewer #2: Yes

6. Review Comments to the Author

Reviewer #1: 1) My previous question regarding distance/speed was: When you say for a people in what speed they can walk you should also provide for what distance. My speed to walk a block is higher than a marathon.

2) I am still not understanding the restriction in use the percentage of predicted of 6MWT. No good answer was provided and more important a proposed equation was shown including BMI, sex, and age - All factors that are controlled using the percentage of predicted in instead of absolute values.

3) By the way, please remove this equation. The design of the study was not adequate to propose it.

4) In conclusion it is stated "It seems appropriate to estimate MWD even for people with little or no schooling". I disagree with this statement considering the correlations obtained (no more than 0.6, which represent a coefficient of explanation lower than 30%). While you can say that it is correlated to MWD, say that it is appropriate to estimate is too much.

Reviewer #2: (No Response)

7. PLOS authors have the option to publish the peer review history of their article (what does this mean?). If published, this will include your full peer review and any attached files.

Reviewer #1: No

Reviewer #2: No

---

## [Author Response · Author response to Decision Letter 2]

15 Nov 2021

Responses to reviewer #1: 

1) My previous question regarding distance/speed was: When you say for a people in what speed they can walk you should also provide for what distance. My speed to walk a block is higher than a marathon.

Thank you for clarifying your question. It is perfectly true that athletes adapt their speed (Walking jogging, running) to the distance that they have to do. In daily life activities and for walking, the difference is very small and the last question about the usual pace is not for running or jogging or sports. This is not what is asked of the patient in the WELSH questionnaire. In the WELSH, patients are asked to estimate only their maximum walking time in the context of daily life activities, for a speed (slow, normal and fast illustrated by a turtle, a human and a rabbit) relative to people of the same age, family or friends. This therefore implicitly incorporates distance as the product of speed and time. The last question relates to usual speed in daily life activities. Indeed asking the patients to define the time (or distance) they can walk for each of possible walking speed does not provide information on whether the person is a slow walkers in his/her usual activities not for any kind of marathon or other unusual distance. We have completed the paragraph of the method section to avoid confusion of reader but would like to kindly underline than we were never reported that patients understood this last item to not relate to their daily routine life (and not sports activities).

2) I am still not understanding the restriction in use the percentage of predicted of 6MWT. No good answer was provided and more important a proposed equation was shown including BMI, sex, and age - All factors that are controlled using the percentage of predicted in instead of absolute values.

We apologize but strongly disagree with the reviewer about the need to compare the WELSH score to an equation. Generally, these equations have their own variability around the mean introducing a new confounding factor in the analysis. Further due to the multiplicity of available equations making any choice would be highly criticized. If dealing with adjusted formulas the major issue becomes the formula themselves and we strongly advocate that using absolute measured values for MWT is the best way to do. Please consider the following equations that were all published for the 6MWT and available in the literature (including for African populations).PMID : 10515400 : PMID: 25053680 : In Africa : PMID: 19472695 PMID: 19041233. We tested the results obtained with the four equations and found dramatically different results.

Unwilling to use a previously published equation, we tried to indirectly do what the reviewer suggests and searched (as a post-hoc analysis) for a formula accounting for gender, age, weight and stature as previous authors did and to try to reach a correlation with the welsh and these parameters to the measured 6MWD as previous authors did. We do understand the reviewer but humbly ask that the reviewer keep in mind that the goal of the study was to have an easy tool for routine use and mental calculation…. which becomes completely impossible if the WELSH was to be adjusted for age/sex/weight /stature. It had been added as a post-hoc analysis after the previous round of correction but according to the next comment, we have now removed it. 

3) By the way, please remove this equation. The design of the study was not adequate to propose it.

The manuscript has been corrected to account for this comment and we have added the point that the step by step analysis to account for sex, BMI and age was a post-hoc analysis and shortly commented this point in the discussion. We have also added a paragraph to underline this as a limitation of our study

4) In conclusion it is stated "It seems appropriate to estimate MWD even for people with little or no schooling". I disagree with this statement considering the correlations obtained (no more than 0.6, which represent a coefficient of explanation lower than 30%). While you can say that it is correlated to MWD, say that it is appropriate to estimate is too much.

We thank the reviewer for this suggestion, the manuscript has been corrected to take this comment into account, and the expression has been removed.

---

## [Decision Letter · Decision Letter 3]

19 Nov 2021

The Walking Estimated Limitation Stated by History (WELSH) visual tool is applicable and accurate to determine walking capacity, even in people with low literacy level.

PONE-D-21-07947R3

Dear Dr. Abraham,

We’re pleased to inform you that your manuscript has been judged scientifically suitable for publication and will be formally accepted for publication once it meets all outstanding technical requirements.

Kind regards,

Sinan Kardeş, M.D.

Academic Editor

PLOS ONE

Additional Editor Comments (optional):

Reviewers' comments:

Reviewer's Responses to Questions

**Comments to the Author**

1. If the authors have adequately addressed your comments raised in a previous round of review and you feel that this manuscript is now acceptable for publication, you may indicate that here to bypass the “Comments to the Author” section, enter your conflict of interest statement in the “Confidential to Editor” section, and submit your "Accept" recommendation.

Reviewer #1: All comments have been addressed

2. Is the manuscript technically sound, and do the data support the conclusions?

Reviewer #1: Yes

3. Has the statistical analysis been performed appropriately and rigorously? 

Reviewer #1: Yes

4. Have the authors made all data underlying the findings in their manuscript fully available?

Reviewer #1: Yes

5. Is the manuscript presented in an intelligible fashion and written in standard English?

Reviewer #1: Yes

6. Review Comments to the Author

Reviewer #1: (No Response)

7. PLOS authors have the option to publish the peer review history of their article (what does this mean?). If published, this will include your full peer review and any attached files.

Reviewer #1: No

---

## [Editor Report · Acceptance letter]

6 Jan 2022

PONE-D-21-07947R3 

The “Walking Estimated Limitation Stated by History” (WELSH) visual tool is applicable and accurate to determine walking capacity, even in people with low literacy level. 

Dear Dr. Abraham:

I'm pleased to inform you that your manuscript has been deemed suitable for publication in PLOS ONE. Congratulations! Your manuscript is now with our production department. 

Kind regards, 

on behalf of

Dr. Sinan Kardeş 

Academic Editor

PLOS ONE